# Assessing Prevalence and Unique Risk Factors of Suicidal Ideation among First-Year University Students in China Using a Unique Multidimensional University Personality Inventor

**DOI:** 10.3390/ijerph191710786

**Published:** 2022-08-30

**Authors:** Ou Wu, Xi Lu, Kee Jiar Yeo, Yunyu Xiao, Paul Yip

**Affiliations:** 1Shulan International Medical College, Zhejiang Shuren University, Hangzhou 310009, China; 2Mental Health Education & Counseling Center, Hangzhou Vocational and Technical College, Hangzhou 314423, China; 3School of Education, University Teknologi, Johor Bahru 81310, Malaysia; 4Department of Population Health Science, Weill Cornell Medicine, NewYork-Presbyterian Hospital, Cornell University, New York, NY 10021, USA; 5Hong Kong Jockey Club Centre for Suicide Research and Prevention, The University of Hong Kong, Hong Kong SAR 999077, China; 6Department of Social Work and Social Administration, Faculty of Social Sciences, The University of Hong Kong, Hong Kong SAR 999077, China

**Keywords:** suicidal ideation, college student, psychological influencing factors, university personality inventory

## Abstract

Background: University students with suicidal ideation are at high risk of suicide, which constitutes a significant social and public health problem in China. However, little is known about the prevalence and associated risk factors of suicidal ideation among first-year university students in China, especially during the COVID-19 pandemic. Objectives: To investigate the prevalence of suicidal ideation and its factors in first-year Chinese university students from a vocational college in Zhejiang during the COVID-19 pandemic. Methods: Using a cluster sampling technique, a university-wide survey was conducted of 686 first-year university students from Hangzhou in March 2020 using University Personality Inventory (UPI). UPI includes an assessment for suicidal ideation and possible risk factors. Suicidal ideation prevalence was calculated for males and females. Univariate analysis and multivariable logistic regression models were conducted, adjusting for age and sex. Analyses were carried out using the SPSS version 22.0 software. Results: The prevalence of 12-month suicidal ideation among first-year university students during March 2020 was 5.2%, and there was no significant difference between males and females (4.8% vs. 6.0%, $x^{2}$ = 0.28, *p* = 0.597). Multivariable logistic regression analysis identified social avoidance (B = 0.78, OR = 2.17, *p* < 0.001) and emotional vulnerability (B = 0.71, OR = 2.02, *p* < 0.001) as positively associated with suicidal ideation. Conclusions: Social avoidance and emotional vulnerabilities are unique factors associated with greater suicidal ideation among first-year university students during the COVID-19 pandemic. UPI serves as a validated tool to screen suicide risks among Chinese university students. Encouraging social engagement and improving emotional regulation skills are promising targets to reduce suicidal ideation among first-year university students.

## 1. Introduction

Suicide is a serious public health problem and is one of the leading causes of death among adolescents and young adults, especially among university students. A meta-analysis reported that the point prevalence of suicidal ideation among Chinese university students ranged between 1.2% and 26.0%, and the prevalence of suicidal ideation among university students in southern China was higher than that in northern China [1]. Understanding the prevalence and risk factors of suicidal ideation is important because it is closely related to suicide attempts and deaths [2,3,4].

Suicidal ideation among first-year university students is of particular importance because of the dynamic developmental, social, and behavioral transitions during their first year at university [5,6,7,8]. Socially, there are changes in residence, family relationships, and peer contexts [5,6,9,10,11]. For most Chinese first-year university students, it is the first time they have started to live in a province other than their place of birth [7,10,12]. Hence, the transition to university involves significant life changes (e.g., increased independence, social demands) and academic challenges (e.g., independent learning) alongside reduced parental support and oversight [13,14] Developmentally, adolescents and young adults have heterogeneous trajectories of suicidal thoughts and behaviors [15]. Greater impulsivity and self-reported aggression, as well as elevated trait anxiety, are the risk factors for suicidal ideation among young adults [15].

Previous studies found that among 16-year-old university students, those with greater tolerance toward suicide, higher family coping, and lower self-esteem were more likely to report suicidal ideation [16]. College entrance may be a strategically well-placed “point of capture” for detecting late adolescents with suicidal thoughts and behaviors [5]. However, a clear epidemiological picture of suicidal thoughts and behaviors among incoming Chinese university students is lacking [5].

Most previous studies relied on patient and hospital records, which may understate the issue of student suicidal ideation [14]. The first twelve months following the onset of suicidal ideation appear crucial, with data from seventeen countries demonstrating that 60% of attempts are made within this period [17]. Nevertheless, previous studies are limited by the focus on psychological risks [14]. However, non-psychological risk factors may be particularly important for Chinese college first-year university students, as they may not talk about their problems due to mental health stigma [18,19]. There is an urgent need to find a customized assessment tool for Chinese university students.

The University Personality Inventory (UPI) was developed in 1966 in Japan and has been adopted as a rapid and effective mental health screening tool among Chinese university students, applied in almost every university in Zhejiang province [20,21]. UPI has been shown to be multidimensional [20,21,22]. Originally, the UPI only concentrated on psychological symptoms, such as depression, anxiety, neuroticism, persecutory beliefs, and obsessive compulsive symptoms [20]. However, these assessments do not help us understand the social and developmental stressors that are relevant to university students specifically [23]. In 2015, a new five-factor structure consisting of domains of physical symptoms, cognitive symptoms, emotional vulnerability, social avoidance, and interpersonal sensitivity of UPI was developed for Chinese university students [21], offering opportunities to understand how the new five-factor domains are associated with suicidal ideation.

During COVID-19, there have been concerns and literature suggesting an increase in social isolation and mental distress, which are potential risk factors for suicidal ideation [24,25]. Special attention will be placed on disparities in suicide prevention across sociodemographic subgroups during the COVID-19 pandemic [26]. For first-year university students, most of their courses have transitioned to an online format, rendering them unable to connect to their classmates and teachers [27]. Those students who felt more connected with other students and teachers were more likely to feel calm and trusting [28]. Previous literature reported the various suicidal ideation situations in disparate Chinese university students, indicating the prevalence range of suicidal ideation from 1.24% to 26.00% [1,29,30,31,32].

However, it is unknown how national prevalent suicidal ideation was among first-year university students in China. Despite recent attention being paid to the alarming rates of suicidal behaviors among university students [5,33,34,35], less evidence is available to understand the potential risks associated with their suicidal ideation during the COVID-19 pandemic. Less research comparatively addressed suicide prevention and early intervention for university students than for primary and secondary school students [36]. This is troubling on the grounds that the college years represent a critical and unique developmental stage characterized by dynamic social role transitions, new living situations, and changing relationships [37,38]. Additionally, it is vital to understand and design college first-year university students specific intervention programs targeting the developmental factors during the transition from adolescence to emerging adulthood [26,39].

This study aimed to investigate the prevalence of suicidal ideation and the potential relationship between the new five-factor domain in the revised UPI and suicidal ideation among Chinese first-year university students during the pandemic period, and present targeting measures.

## 2. Materials and Methods

### 2.1. Data Collection and Sample

The study was held in a higher vocational college (similar to a community college) located in Xiaoshan district, Hangzhou, Zhejiang. Hangzhou is the capital and most populous city of Zhejiang Province, People’s Republic of China [40,41,42]. An online questionnaire was distributed to all of the first-year students in this college. This college is somehow like a community college, and is partially guided by Zhejiang Open University, which governs nearly eighty similar colleges all around the province. There are two types of students. One is a full-time student (completes a 3-year course of study to get an associate degree, after which the qualified student could either take an entry exam for further study to get a bachelor’s degree or enter employment); the other is a part-time student who usually has employment and already has an associate degree, and chooses to take courses at the weekend for further study for a bachelor degree. Using cluster sampling, we selected one higher vocational college (Xiaoshan college) to invite 686 first-year students to participate in this investigation (effective response rate = 100.00%). Our study sample is full-time students who just graduated from high school and were of similar age. Our sample is unique, as they were the first batch of first-year students entering college during the COVID-19 pandemic. Using a cluster sampling technique, a total of 686 (248 women, 438 men; ages 17 to 19) of these first-year university students were selected. The study was approved by the Human Research Ethics Committee of the vocational college. Informed written consent was obtained from participants before the study commenced.

During the first month of students’ school entry (March 2020), participants completed the UPI with demographic information (age, sex). Students with a UPI total sum score above 20 or those who responded “yes” to item 25 (“Have an idea of wanting to die”) were identified as a risk group and were referred to a mental health professional [20,21].

### 2.2. Measures

The UPI is a 60-item self-report measure assessing whether an individual experienced one or more mental health symptoms during the past year [20]. For each item, a score of 1 is given for “Yes”, and 0 was given for “No”. The 56 items describe whether an individual experienced one or more mental health symptoms, except for the “lie” scales (items 5, 20, 35, and 50). (The full version of the UPI is available in the Appendix A).

Suicidal ideation was measured by one item response to “Have an idea of wanting to die?” Participants responding “yes” were considered to have suicidal ideation. 

The revised UPI includes five main factors tailored to assess physical, social, cognitive, interpersonal, and emotional health for Chinese university students (Table 1): (1) Physical symptoms (items 1, 2, 3, 17, 18, 46, 48 and 49); (2) social avoidance (items 10, 11, 41 and 43); (3) cognitive symptoms (items 29, 30, 38 and 39); (4) interpersonal sensitivity (items 57, 58); and (5) emotional vulnerability (items 6,15, 21, 24, 28 and 60). Factors were equal to the summed scores of the respective items based on the previously validated scoring guide [21]. For details, see Table 1.

### 2.3. Data Analyses

Descriptive statistics (frequencies, percentages, means, and standard deviations) were used to report socio-demographic data and the prevalence of suicidal ideation. Univariate analyses (Chi-square test, rank-sum test, or binary logistic regression models) were implemented to assess the differences of five factors by suicidal ideation status. Multivariate logistic regression models were then fitted to explore associations between five UPI factors and suicidal ideation, controlling for the covariates (sex, age, major-, e.g., academic subject). A significance level of *p* < 0.05 was applied for all significance tests. Associations were reported as Odds Ratios (OR) and 95% confidence intervals (CI). All analyses were performed using SPSS for Windows V22.0 (IBM Corp., Armonk, NY, USA). There were no missing values (response rate: 100%).

## 3. Results

### 3.1. Sample Characteristics

Table 2 presents the sample characteristics. There were more male university students (*n* = 438, 63.85%) than females (*n* = 248, 36.15%). On average, students were 17.79 years old. Most students had network engineering, business administration, and preschool education majors.

### 3.2. Prevalence of Suicidal Ideation

The 12-month prevalence of suicidal ideation in the total sample was 5.2% (4.8% in men, 6.0% in women). There was no significant sex difference in suicidal ideation prevalence ($x^{2}$ = 0.28, *p* = 0.597).

### 3.3. Associations between UPI Factors and Suicidal Ideation

For all UPI factors, students with suicidal ideation had significantly higher scores than those without suicidal ideation (See Table 3).

Results of multivariable regression (Table 4) showed significant differences in social avoidance and emotional vulnerability to suicidal ideation, but not the rest of the UPI factors. Specifically, students with social avoidance were 2.17 times more likely to think about suicide than those without. Similarly, students with emotional vulnerability had 2.03 greater odds of suicidal ideation than those without.

## 4. Discussion

In this study, we found suicidal ideation prevalence was 5.2% among the 686 first-year university students from the higher vocational college in Hangzhou, Eastern China. Such a rate is similar to an earlier study during COVID-19 among 2700 higher vocational college second-year university students in China (5.4%) [32], but lower than that in the US during the COVID-19 quarantine period (20%) in early 2020 [43,44].

Suicidal ideation prevalence in China increased during the COVID-19 period compared with the pre-pandemic period in China (1.52% in Shandong), based on a prior study from April 2018 to October 2019 [30]. Suicidal ideation prevalence may vary with alteration of the years and areas. Research found that the suicidal ideation prevalence of Chinese higher vocational college first-year university students was 1.05% in 2008 in Jiangsu province, Eastern China [45]. Another research study found suicidal ideation prevalence was 2.63% in Chinese private higher vocational college first-year university students in 2009 in Anhui province, Eastern China [46]. The suicidal ideation prevalence among higher vocational college first-year university students mentioned above was lower than that found in this study.

Only social avoidance and emotional vulnerability were significant in the multivariable logistic regression after adjusting for covariates, while physical symptoms, cognitive symptoms, and interpersonal sensitivity were correlated with suicidal ideation. Social avoidance is commonly defined as the desire to escape or actually avoid being with, talking to, or interacting with others for any reason, i.e., the unwillingness to connect with the outside world in this study [47,48]. Previous literature also presented that social avoidance in young patients is a clinically worrisome phenomenon that characterizes impending schizophrenia, but that also constitutes a core phenomenon in avoidant personality disorder, schizoid personality disorder, and autism spectrum disorder [24,49]. It was demonstrated there was a causal pathway from social network strain (due to adverse childhood experiences) to suicidal ideation among middle-aged adults [24]. Social networks during adolescence influenced the odds of belonging to distinct suicidal trajectories [50]. Family cohesion protects youth from being in high-risk developmental courses of suicidal behaviors [50]. Social networks, especially quality of interactions, may improve suicide risk screening among university students [50].

Social restriction implemented as the quarantine measure during the COVID-19 pandemic and the perceived social isolation [51] has contributed to significant psychological distress [52]. The COVID-19 pandemic affected different aspects of daily life and mental well-being, making it difficult to reduce its influence on a single factor such as social avoidance [53,54]. It was found that social avoidance is a social risk of youth development [55]. Social avoidance increases loneliness, and so we need to build connectedness for students [56].

Emotional vulnerability is commonly defined as the willingness to acknowledge one’s emotions, especially painful ones such as shame, sadness, anxiety, insecurity, etc. [57,58]. It is also been defined as “uncertainty, risk, and emotional exposure”, and is felt as anxiety about being rejected, shamed, or judged as inadequate, i.e., the manifestations of negative emotion in this study. Previous research suggests that problems with emotion regulation may be implicated in suicidality [59,60]. Early problems in learning to read and spell are related to motivational–emotional vulnerability in learning situations in the school context [61]. Self-reported ability to manage self-relevant emotions was negatively associated with suicidal ideation among university students [62]. Additionally, endorsement of fewer adaptive responses to negative emotions, as well as more maladaptive responses, increased the odds of suicidal ideation, plans, and attempts in children [63,64]. Emotional vulnerability can be related to personality, adverse childhood experiences, and also recent stress [24,65,66,67]. This study quantified the effect of risk elements in emotional vulnerability factors on suicidal ideation in a higher vocational college.

Thirdly, the most important thing is that under the five-factor frame of UPI, the change of every item score in the five factors was found to have a vital significant relationship with the risk of suicidal ideation prevalence, while, when these items were considered solely, respectively, these vital significant relationships could not be found. That may be the fundamental meaning of the five-factor frame of UPI. Thus, it is known to all that when a first-year university student has one of the following situations, “Do not like meeting others”, “Feel that I am not myself”, “Lack faith in others”, “Unwilling to associate with others”, “Full of dissatisfaction and complaints”, “Over-uneven in emotion”, “Intolerance”, “Irritable”, “Lack of patience”, and “Sensitive emotions”, this student’s suicidal ideation risk will increase. Additionally, these phenomena were not fully illustrated by other researchers. Given this wide and varied spectrum of related factors, a logical next step would be to develop and examine the effectiveness of targeted interventions that are “personalized” to not only the suicidal ideation, but also its associated risk factors.

Our study is the first to demonstrate the utility of the revised UPI in understanding the risks of suicidal ideation among university students in China. In particular, by adding the five factors relevant to living experiences tailored to Chinese students, we were able to find two important factors—social avoidance and emotional vulnerability—as potential risk factors of suicidal ideation amid the COVID-19 pandemic. While previous studies mainly focused on psychological distress, social and emotional regulations may be unique protective factors for first-year university students [68]. In China, some Promoting Alternative Thinking Strategies (PATHS) Curriculum interventions help students improve their emotional understanding, emotion regulation, and prosocial behavior [69]. Additionally, child routines were shown to mediate the relationship between parenting and social–emotional development in China, which have important implications for research and practice aimed at enhancing social–emotional outcomes [70]. Our findings suggest that social avoidance and emotional vulnerability are important and unique risk factors for suicidal thoughts among university students in China, which has important implications for research and practice. Previous studies demonstrate that interpersonal belongingness [71,72], anhedonia, and type D personality [73] are potentially strong predictors in medical students. School-based suicide prevention can target building belongingness, student–teacher connectedness, more engaging school climates, and emotional well-being surveillance to screen high-risk students.

Some limitations of this study exist. This is a cross-sectional study and lacks comparison with pre-COVID time periods. Moreover, it was limited by COVID-19 epidemic prevention and control restrictions, and the differences in the entrance to the university in different regions. We were not able to control for further covariates; for example, we do not have data on the living arrangements of students due to the lack of information in the survey. Only one college was investigated in this study, and all respondents were first-year university students, so there may be some limitations in the representativeness of the sample and extrapolation of the results.

## 5. Conclusions

In conclusion, the study revealed that for first-year university students amid the COVID-19 pandemic, the factors of physical symptoms, cognitive symptoms, and interpersonal sensitivity may work through the factors of social avoidance and emotional vulnerability. If any item of the social avoidance factor or of the emotional vulnerability factor, comes true, the likelihood of suicidal ideation will increase more than two times for university students. With the five-factor structure UPI, more associated risk factors can be found. Additionally, these findings contribute to providing timely intervention measures to reduce suicidal ideation and reduce the risk of suicide in this college student population.

## Figures and Tables

**Table 1 ijerph-19-10786-t001:** Five factors in revised UPI tailored to Chinese students.

Five Factors	Descriptions	Question Items
1. Physical symptoms	Feeling bad physically	Poor appetite; Feel sick, stomachache; Easily have diarrhea or constipation; Headache; Ache in neck and shoulder; Physically exhausted; Dizzy when I stand up; Have ever lost consciousness, cramp
2. Social avoidance	Unwillingness to connect with the outside world	Do not like meeting others; Feel that I am not myself; Lack faith in others; Unwilling to associate with others
3. Cognitive symptoms	Cognitive decline and negative self-image	Lack of judgment; Too dependent on others; Lack of confidence; Irresolute about anything
4. Interpersonal sensitivity	Being sensitive to others’ feeling and perceptions	Wary of others; Care about others’ gaze;
5. Emotional vulnerability	Manifestations of negative emotion	Full of dissatisfaction and complaints; Over-uneven in emotion; Intolerance; Irritable; Lack of patience; Sensitive emotions

**Table 2 ijerph-19-10786-t002:** Sample characteristics.

	Study Sample (*n* = 686)
	*n* (%)
Sex	
Female	248 (36.15)
Male	438 (63.85)
Age, mean (SD)	17.79 (0.78)
Major	
Business Administration	113 (16.47)
Accounting	76 (11.08)
Mechanical and Electronic Engineering	100 (14.58)
Building Engineering/Constructional Engineering	70 (10.20)
Building Economic Management	17 (2.48)
Network Engineering	131 (19.10)
Logistics	72 (10.50)
Preschool Education	107 (15.60)

**Table 3 ijerph-19-10786-t003:** The differences between suicidal ideation and non-suicidal ideation of the scores of five factors.

Five Factors	Non-Suicidal Ideation	Suicidal Ideation	*p* Value
	(Mean ± SD)	(Mean ± SD)	
F1 (Physical symptoms)	0.76 ± 1.36	3.89 ± 2.79	<0.001
F2 (Social avoidance)	0.33 ± 0.81	2.64 ± 1.40	<0.001
F3 (Cognitive symptoms)	0.61 ± 1.10	2.64 ± 1.47	<0.001
F4 (Interpersonal sensitivity)	0.46 ± 0.76	1.28 ± 0.88	<0.001
F5 (Emotional vulnerability)	0.70 ± 1.28	3.89 ± 1.56	<0.001

**Table 4 ijerph-19-10786-t004:** Multivariable logistic regression models for suicidal ideation with the five factors.

Factors	B	Wald χ^2^	*p* Value	Odds Ratio	95% Confidence Interval
F1 score (Physical symptoms)	0.17	2.21	0.137	1.18	0.95–1.47
F2 score (Social avoidance)	0.78	15.12	<0.001	2.17	1.47–3.21
F3 score (Cognitive symptoms)	−0.21	0.67	0.408	0.81	0.49–1.34
F4 score (Interpersonal sensitivity)	−0.51	2.19	0.139	0.6	0.31–1.18
F5 score (Emotional vulnerability)	0.71	12.34	<0.001	2.02	1.37–3.00

## Data Availability

The datasets generated and analyzed during the current study are not publicly available due to funding policy but are available from the corresponding author on reasonable request.

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
