# Peer review of "Assessing Prevalence and Unique Risk Factors of Suicidal Ideation among First-Year University Students in China Using a Unique Multidimensional University Personality Inventor"

_ijerph, 2022, doi:10.3390/ijerph191710786_

Round 1

Reviewer 1 Report

It was  a real pleasure to review this well written manuscript addressing one of the most crucial topics in mental health literature amidst recent pandemic, ie suicidality in youths. The research design, the sample size, the data analysis and above all the clarity of result presentation and findings discussion make this paper suitable for publication in a peer reviewed journal. Highlighting social avoidance and emotional vulnerability as very important risk factors for suicidal thoughts among youths has important implication for research and practice. I hope that the journal will reach a positive decision for this manuscript. 

Author Response

Response: Thank you for reviewing our manuscript. We provided substantial details about the research design, the sample size, and the data analysis, together with the results and discussion in the revised version. 

In the discussion, we highlighted social avoidance and emotional vulnerability as very important risk factors for suicidal thoughts among youths. 

Reviewer 2 Report

Thank you for the invitation to evaluate the article

The article addresses a theme of significant relevance to the field of public health and requires minor revisions to have the quality necessary for publication. 

Regarding the place where the study was conducted, it would be important to provide more information so that it becomes evident why it was chosen. 

Authors should include the calculation for sample size. 

The number of males was much higher than females. Considering that suicidal intention and suicide have different behavior between genders, shouldn't the sample have selected an equivalent number between genders?

In the results it is suggested to include a brief characterization of the population that was studied. 

In the first paragraph of the results, inform again the number of observations of the study and present the relative frequencies accompanied by the absolute ones. 

The results would be much more interesting if they included an analysis according to gender and age. 

Author Response

Thank you for the invitation to evaluate the article

Response: Thank you 

The article addresses a theme of significant relevance to the field of public health and requires minor revisions to have the quality necessary for publication. 

Response: Thank you 

  1. Regarding the place where the study was conducted, it would be important to provide more information so that it becomes evident why it was chosen. 

Response: Thank you. We provided where the study was conducted.

  1. Authors should include the calculation for sample size. 

Response: Thank you. We provided the sample size.

  1. The number of males was much higher than females. Considering that suicidal intention and suicide have different behavior between genders, shouldn't the sample have selected an equivalent number between genders?

Response: Thank you. We revised and provided a Table (new Table 2) for clarification.  

  1. In the results, it is suggested to include a brief characterization of the population that was studied. 

Response: Thank you. We provided a brief characterization of the population that was studied. 

  1. In the first paragraph of the results, inform the number of observations of the study again and present the relative frequencies accompanied by the absolute ones. 

Response: Thank you. We provided this information. 

  1. The results would be much more interesting if they included an analysis according to gender and age. 

Response: This is a great suggestion, which we propose to do in the next step (and written in the future direction).

Reviewer 3 Report

Interesting article. The authors must discuss the relationship between social avoidance and emotional vulnerability and similar risk factors of suicidal ideations: (1) perceived burdensomeness and thwarted belongingness of the interpersonal theory of suicide: see Joiner et al, 2021, Preventive medicine, 152, 106453; Solibieda et al, 2021, 18, 11526; (2) anhedonia and type D of personality: Loas et al , Plos one, 2019, 14, 6, e02117841

Author Response

Response: Thank you. We added this information and cited the reference in our discussion (page 7).